# Cryptococcal Immune Reconstitution Inflammatory Syndrome: From Clinical Studies to Animal Experiments

**DOI:** 10.3390/microorganisms10122419

**Published:** 2022-12-07

**Authors:** Zoe W. Shi, Yanli Chen, Krystal M. Ogoke, Ashley B. Strickland, Meiqing Shi

**Affiliations:** 1Division of Immunology, Virginia-Maryland College of Veterinary Medicine and Maryland Pathogen Research Institute, University of Maryland, College Park, MD 20742, USA; 2Department of Chemistry and Biochemistry, University of Maryland, College Park, MD 20742, USA

**Keywords:** fungus, *Cryptococcus neoformans*, HIV/AIDS, antiretroviral therapy (ART), fungal pathogenesis, cryptococcosis, cryptococcal meningitis, meningoencephalitis, inflammation, IRIS, PIIRS, CSF, CNS, central nervous system, brain, CD4^+^ T cells, microglia, monocytes, macrophages, T cells, NK cells, cytokines, chemokines, IFN-γ, TNF-α, biomarkers

## Abstract

*Cryptococcus neoformans* is an encapsulated pathogenic fungus that initially infects the lung but can migrate to the central nervous system (CNS), resulting in meningoencephalitis. The organism causes the CNS infection primarily in immunocompromised individuals including HIV/AIDS patients, but also, rarely, in immunocompetent individuals. In HIV/AIDS patients, limited inflammation in the CNS, due to impaired cellular immunity, cannot efficiently clear a *C. neoformans* infection. Antiretroviral therapy (ART) can rapidly restore cellular immunity in HIV/AIDS patients. Paradoxically, ART induces an exaggerated inflammatory response, termed immune reconstitution inflammatory syndrome (IRIS), in some HIV/AIDS patients co-infected with *C. neoformans*. A similar excessive inflammation, referred to as post-infectious inflammatory response syndrome (PIIRS), is also frequently seen in previously healthy individuals suffering from cryptococcal meningoencephalitis. Cryptococcal IRIS and PIIRS are life-threatening complications that kill up to one-third of affected people. In this review, we summarize the inflammatory responses in the CNS during HIV-associated cryptococcal meningoencephalitis. We overview the current understanding of cryptococcal IRIS developed in HIV/AIDS patients and cryptococcal PIIRS occurring in HIV-uninfected individuals. We also describe currently available animal models that closely mimic aspects of cryptococcal IRIS observed in HIV/AIDS patients.

## 1. Introduction

*Cryptococcus neoformans* is one of the most important human pathogenic fungal species [1,2,3]. The fungus exists in the environment with global distribution and humans become infected by inhaling the fungus [1,2,3]. Although the infection starts in the lung, the fungus can escape from the infected lung and disseminate into the central nervous system (CNS) via the bloodstream, particularly in immunocompromised individuals including HIV/AIDS patients, leading to fatal meningoencephalitis [4,5,6]. In addition to affecting HIV/AIDS patients, cryptococcal meningoencephalitis occurs in HIV-negative individuals such as organ transplant patients, even in immunocompetent individuals [7,8]. The number of deaths caused by this disease remains unacceptably high. Currently, there are 223,100 incident cases of cryptococcal meningoencephalitis diagnosed each year, and it is estimated that 181,100 people die from this disease globally each year [9]. Thus, cryptococcal meningoencephalitis is a serious threat to human health.

Th1 immune responses, characterized by IFN-γ production and classic activation of macrophages, is required for the clearance of a cryptococcal infection, whereas animal studies suggested that a Th2 immune response, characterized by secretions of IL-4, IL-5 and IL-13 and alternative activation of macrophages worsens the disease [10,11,12,13]. The susceptibility of HIV/AIDS patients to *C. neoformans* infection is attributed to impaired cellular immune responses, particularly the quantitative and qualitative defects of CD4^+^ T cells [14]. Clinical studies demonstrated that higher levels of IFN-γ and TNF-α in cerebrospinal fluid (CSF) and/or serum were correlated to a better fungal clearance and enhanced survival of HIV/AIDS patients co-infected with *C. neoformans* [15,16]. Adjunctive therapy with recombinant IFN-γ resulted in enhanced fungal clearance in the CSF and improvement of cryptococcal meningoencephalitis, confirming the important role of IFN-γ in fighting *C. neoformans* in the CNS [17,18]. However, *C. neoformans* possesses multiple virulence factors, mainly the polysaccharide capsule [19,20], that is capable of modulating host immune responses [21,22].

Antiretroviral therapy (ART) represents one of the major advances in treatment of HIV infections. Following ART, the rapid recovery of the immune responses in HIV/AIDS patients with cryptococcal meningoencephalitis led to immune reconstitution inflammatory syndrome (IRIS), that occurred in 10% to 45% of HIV/AIDS patients with cryptococcal meningoencephalitis [23,24,25,26,27,28], with a mortality of 33% to 66% [23,25,26,29]. Cryptococcal IRIS is an exaggerated inflammatory response in the CNS, characterized by a cytokine storm with a predominant Th1/IFN-γ response, activation of coagulation, and neuronal cell activation and damage [24,30]. Poor inflammation and higher fungal burdens in the CNS before ART is predictive of subsequent IRIS after initiation of ART [7,30,31,32]. Likewise, a lethal inflammatory response, termed post-infectious inflammatory response syndrome (PIIRS), has also been observed in the CNS of HIV-uninfected previously healthy individuals suffering from cryptococcal meningoencephalitis [7,8,33]. PIIRS is characterized by a remarked accumulation of immune cells including T cells and monocytes and IFN-γ secretions, accompanied by neurological damage, despite effective fungal control [8]. Studies using animal models have confirmed that cryptococcal IRIS and PIIRS was induced mainly by IFN-γ-producing CD4^+^ T cells [34,35,36].

## 2. Inflammatory Responses in the CNS during HIV-Associated Cryptococcal Meningoencephalitis

Compared to healthy individuals, HIV/AIDS patients with cryptococcal meningoencephalitis displayed higher CSF levels of IFN-γ, TNF-α, IL-6, IL-7, IL-8, IL-10, IL-12p40, IL-15, IL-18, and CCL2 [37]. They also had higher levels of CSF IL-12p40 and IL-17A compared with HIV/AIDS patients without co-infection of *C. neoformans*, suggesting inflammation of the CNS during cryptococcal meningoencephalitis [37]. Inflammatory responses are thought to play a prominent role in the eradication of cryptococcal infections in the CNS [38]. By analyzing clinical samples of 62 patients, Siddiqui et al. revealed that there were significantly higher levels of CSF pro-inflammatory cytokines IFN-γ, TNF-α, IL-6 and IL-8 in survivors of HIV patients with cryptococcal co-infection compared to non-survivors and that higher levels of these cytokines were correlated with reduced fungal burdens, supporting that these cytokines promote fungal clearance [16]. These cytokines were likely produced by activated resident cells, as there was no correlation between CSF leukocyte counts and levels of these cytokines [16]. In addition, analysis of human brain autopsy tissues of HIV/AIDS patients with cryptococcal meningoencephalitis revealed limited infiltration of macrophages and lymphocytes in the brain parenchyma [39]. A significant concentration of chemokines MIP-1β and MCP-1, and cytokines G-CSF and GM-CSF were detected in the CSF, but these inflammatory mediators were not associated with clinical outcome [16]. In support of the protective role of proinflammatory cytokines, a later study also showed that HIV/AIDS patients who have survived cryptococcal meningoencephalitis exhibited higher CSF levels of IL-8, IL-12p40, IL-17A, TNF-α, INF-γ and serum TNF-α compared to non-survivors [40]. In line with observations of CSF cytokines, it was reported that higher secretions of IFN-γ and TNF-α by peripheral CD4^+^ T cells were associated with improved survival of HIV/AIDS patients with cryptococcal meningoencephalitis [41].

By analyzing CSF samples from 90 HIV/AIDS patients with cryptococcal meningoencephalitis, Jarvis et al. demonstrated that a robust immune response consisting primarily of Th1, Th2, and Th17-type cytokines was correlated with more rapid fungal clearance from CSF and improved survival during treatment of HIV-associated cryptococcal meningoencephalitis [15]. The major components of the protective immune responses were IL-6 and IFN-γ, the signature cytokines of a Th1 response [15]. In addition, Th2-type (IL-4 and IL-10) and Th17-type (IL-17) cytokines contributed, to a lesser extent, to protection against cryptococcal infection [15]. The protective roles of Th1 and Th17 responses in HIV-associated cryptococcal meningoencephalitis are consistent with findings from animal studies [42,43,44,45,46,47,48,49]. Of note, it has been shown that IFN-γ is required to activate microglia, promoting clearance of cryptococcal infection in a murine model [50,51]. In mouse studies, Th2 responses and alternative activation of macrophages were associated with exacerbation of disease during *C. neoformans* infection [52,53,54,55]. Interestingly, CSF levels of Th2 cytokines (IL-4 and IL-10) were closely correlated with the IFN-γ level, and both cytokines were associated with better clearance of *C. neoformans* and reduced mortality, rather than worsening of the disease, reflecting differences in immune responses to *C. neoformans* infections between humans and animals [15]. Of note, arginase activity, an indicator of alternative activation of macrophages [44,56], was not linked to disease severity of cryptococcal meningoencephalitis [15]. Moreover, based on flow cytometry analysis of blood samples from 57 HIV/AIDS patients with cryptococcal meningoencephalitis, a paucity of CD4^+^ and CD8^+^ T cells, and lower IL-6, G-CSF, and IL-5 secretions, rather than alternative macrophage activation, was associated with inadequate fungal clearance and severe disease [57].

In support of the beneficial role of inflammatory response in HIV-associated cryptococcal meningoencephalitis, it has been shown that addition of IFN-γ to standard treatment significantly enhanced the rate of clearance of *C. neoformans* infection from the CSF [17,18]; by contrast, adjunctive therapy with immune-suppressing corticosteroids did not reduce the mortality of AIDS patients co-infected with *C. neoformans*, but was associated with slower fungal clearance in CSF [58]. Obviously, impaired cellular immune responses in HIV/AIDS patients facilitate fungal growth, leading to fungus-mediated brain damage during cryptococcal meningoencephalitis [59].

## 3. Cryptococcal IRIS in HIV/AIDS Patients

Although inflammation in the CNS is essential for clearance of a cryptococcal infection in HIV/AIDS patients [38], paradoxically there is increasing evidence linking brain damage and mortality of HIV/AIDS patients to inflammation in the CNS [7,59]. In particular, after initiation of ART, 10% to 45% of HIV/AIDS patients with cryptococcal meningoencephalitis developed IRIS [23,24,25,26,27,28], a lethal inflammatory response that complicates the recovery from immunodeficiency [60]. In addition, some transplant recipients developed a clinical syndrome similar to IRIS after reductions in immunoconditioning [61,62]. Due to the closed compartment of the brain within the skull, the excessive inflammation and resulting cerebral edema often had a serious outcome during IRIS [7].

IRIS often occurred in HIV/AIDS patients with poor baseline inflammatory responses and uncontrolled cryptococcal growth in the CNS [32,63,64] (Table 1). At ART initiation, patients with high fungal burdens [26,64], low CSF protein levels [63,64], low peripheral CD4^+^ T cell counts [15,24,26] and CSF leukocyte counts [15,63], and reduced CSF levels of IFN-γ, TNF-α, IL-6, and IL-8 [63] had a higher risk for developing IRIS. A lower secretion of IFN-γ by peripheral CD4^+^ T cells and a lower activation of the peripheral CD4^+^ T cells prior to ART were risk factors for IRIS [65]. A lower CD4^+^ T cell increase during ART was also a strong predictor of IRIS [64]. Following in vitro stimulation of peripheral blood mononuclear cells (PBMCs) by cryptococcal glucuronoxylomannan (GXM), lower frequencies of CD4^+^ and CD8^+^ T cell memory subsets expressing IL-2, IFN-γ, or IL-17 were detected prior to ART among patients who subsequently developed IRIS [66]. Patients who later developed IRIS exhibited higher proportions of CXCR3^+^CCR5^+^CD8^+^ T cells in the CSF compared with blood (i.e., a higher CSF-blood ratio of CXCR3^+^CCR5^+^CD8^+^ T cells) at ART initiation, suggesting that a higher CD8^+^ T cell response may predispose patients to IRIS [67]. Interestingly, a higher CSF-blood ratio of CXCL10, CCL2 and CCL3 was also observed at ART initiation among those who later developed IRIS [67]. In addition, CCL2/CXCL10 and CCL3/CXCL10 CSF ratios were increased at ART initiation in patients who developed IRIS [67]. Given that CCL2 and CCL3 are chemotactic for monocyte/macrophages, whereas CXCL10 is chemotactic for CXCR3^+^ T cells, this observation may suggest that monocyte/macrophage responses to *C. neoformans* in the CSF prior to ART is linked to increased risk of IRIS [67]. In line with this hypothesis, Jarvis et al. reported that higher CSF levels of chemokines MCP-1 (CCL2), MIP-1α (CCL3) and the cytokine GM-CSF at ART initiation were associated with subsequent cryptococcal IRIS [15]; arguably, these mediators promote inflammation by attracting monocyte migration to the CNS [15]. In this context, a higher frequency of peripheral monocytes secreting TNF-α and IL-6 was detected at baseline following ex vivo stimulation with IFN-γ among those who developed IRIS [68].

By analyzing blood samples from 101 HIV/AIDS patients with cryptococcal meningoencephalitis, Boulware et al. evaluated the serum biomarkers in IRIS [24]. They found that patients who developed IRIS had four-fold higher levels of medium serum cryptococcal antigen prior to ART [24], which supported earlier studies showing that a high serum level of cryptococcal antigen was predictive of IRIS [69,70]. Patients with higher serum levels of IL-4 and IL-17 prior to ART had a higher frequency to develop subsequent IRIS, which suggested that there was a closed correlation between Th2/Th17 responses and development of IRIS [24]. A higher serum level of C-reactive protein (CRP) before ART was also associated with increased risk of subsequent IRIS [24]. In addition, lower levels of TNF-α, G-CSF, GM-CSF, and vascular endothelial growth factor (VEGF) in the serum prior to ART were predictive of subsequent IRIS. After initiation of ART, increasing serum levels of CRP, D-dimer, IL-1RA, IL-6, IL-7, IL-13, or G-CSF correlated with a higher risk for IRIS development [24]. Recently, by analyzing whole blood transcriptomic profiles of cryptococcal patients, Vlasova-St Louis et al. reported that low expression of interferon-inducible genes and high expression of transcripts that encode granulocyte-dependent proinflammatory response molecules were predictive of subsequent IRIS [71]. More recently, it has been shown that patients who developed cryptococcal IRIS displayed significantly lower plasma levels of IgM, Lam-binding IgM (Lam-IgM), Lam-IgG, and cryptococcal GXM specific IgM prior to ART initiation, compared to those who did not develop IRIS, demonstrating an association between poor B cell function and IRIS development [72].

IRIS development is characterized by excessive inflammation in the CNS [59] (Table 2). Compared to non-IRIS patients, IRIS patients had greater migration of CD4^+^ T cells [23,73] and intermediate monocytes in CSF [73], increased programed death ligand 1 (PD-L1) expression on NK cells [73], and increased macrophage/microglial activation as evidenced by enhanced CSF concentrations of macrophage activation markers (soluble CD14 and CD163) after ART initiation [74], associated with declining fungal burdens [73]. A recent study showed that there were higher frequencies of activated monocytes (CD14^+^CD86^+^ or CD14^+^HLA-DR^+^) in the peripheral blood of IRIS patients compared with non-IRIS patients [75]. Accordingly, neuropathological examination demonstrated that there were demyelinating lesions in the brain with marked infiltration of macrophages and T lymphocytes in the parenchyma and perivascular space [76]. CSF levels of IFN-γ, TNF-α, G-CSF, VEGF, and eotaxin (CCL11) were significantly elevated in cryptococcal IRIS patients [63]. At the time of IRIS onset, serum levels of multiple proinflammatory cytokines were increased; among them, IL-6 and CRP were most frequently elevated [24]. It is known that IL-6 stimulated CRP secretion in the liver and that IL-6 was primarily secreted by activated macrophages during IRIS [77,78]. Compared to non-IRIS patients, IRIS patients exhibited a significantly higher plasma level of IFN-γ, which may play a role in IRIS pathology [79]. In addition, in vitro stimulation of PBMCs by cryptococcal GXM elicited higher frequencies of IL-2^+^/IL-17^+^ CD4^+^ T cells in patients with IRIS [66]. Interestingly, the expression of anti-inflammatory cytokine IL-10 was also enhanced in IRIS patients compared to non-IRIS patients, although the difference did not reach statistical significance; there was a significant correlation between IL-10, IL-6, and IFN-γ, suggesting that the excessive inflammation that characterizes IRIS occurs first, followed by a compensatory secretion of IL-10 [79]. One of the major cellular sources of IL-10 is Foxp3^+^CD4^+^ Treg cells [80]. In this context, a high frequency of Foxp3^+^CD4^+^ Treg cells was detected in cryptococcal IRIS patients [81,82]. It is noteworthy that tuberculosis-IRIS patients also showed significantly higher concentration of IL-10 in the serum, compared to non-IRIS patients [83]. Given the excessive inflammation in IRIS and the anti-inflammatory function of IL-10, the enhanced secretion of IL-10 may represent a compensatory anti-inflammatory response.

Due to impaired cellular immune responses, HIV/AIDS patients are more susceptible to *C. neoformans* infection and the organisms proliferate in the CNS, causing microbe-mediated brain damage [59]. Following ART, abundant cryptococcal antigens drive IRIS characterized by a substantial recruitment of immune cells to the CNS and excessive inflammation, leading to fungal clearance but at a cost of host-mediated brain damage [59]. Thus, it is crucial to maintain a balance between protective immunity and immunopathology during treatment of HIV-associated cryptococcal meningoencephalitis [59].

## 4. Cryptococcal PIIRS in HIV-Negative Patients

In contrast to the reduced incidence of HIV-associated cryptococcal meningoencephalitis [9,84], the incidence of non-HIV-infected cryptococcal meningoencephalitis is increasing, particularly in developed countries [85]. Non-HIV-infected cryptococcal meningoencephalitis accounts for about one third of all cases of the illness in the USA with up to a 30% death rate despite proper treatment [85,86,87]. There are two categories of non-HIV-infected patients with cryptococcal meningoencephalitis: (1) patients with immune suppression by immunotherapy such as solid organ transplant patients; (2) previously healthy, apparently immunocompetent individuals [7]. For those previously healthy individuals with cryptococcal meningoencephalitis, antifungal therapy did not improve the clinical symptoms but induced clinical deterioration despite effective control of fungal growth in the brain [7,8,33,88,89]. In contrast, adjunctive treatment with corticosteroids can improve the disease [88], suggesting that an aberrant immune response, similar to IRIS, occurred in the CNS of those HIV-uninfected previously healthy patients, termed PIIRS [7,89].

By analyzing CSF and blood samples from 17 previously healthy patients with severe cryptococcal CNS disease (s-CNS) and 6 patients with previous fungal exposure leading to non-CNS disease as well as 11 healthy donors, Panackal et al. demonstrated that there was a significant increase in absolute counts of CSF innate and adaptive immune cells including activated HLA-DR^+^CD4^+^ and CD8^+^ T cells, B cells, cytotoxic (CD56^dim^) and immunoregulatory (CD56^bright^) NK cells, monocytes, and myeloid dendritic cells (MyDCs) in s-CNS patients, compared to non-CNS infected patients and health donors [8]. The increased CSF immune cells correlated with a more than 10-fold higher CSF neurofilament light chains (NFL), a biomarker of axonal damage in neurological diseases [90,91], suggesting a neurological damage mediated by host inflammation [8]. Ex vivo studies by co-culturing T cells from CSF or blood with cryptococcal antigen-loaded DCs demonstrated a compartmentalized Th1 response with enhanced secretions of IFN-γ by CD4^+^ and CD8^+^ T cells in s-CNS patients [8]. Moreover, s-CNS patients displayed significantly higher CSF levels of cytokines IFN-γ, IL-6, IL-18, IL-10, IL-13 and chemokines CXCL10, MCP-1 (CCL2), MIP-1α(CCL3), MCP-3 (CCL-7), and MIP-3β (CCL7), but significantly lower CSF levels of Th2 cytokine IL-4, compared to non-CNS patients and/or health donors, further demonstrating a robust Th1 response and active inflammation in the CNS of s-CNS patients [8] (Table 3).

Immunohistochemistry staining of brain biopsy and autopsy specimens from s-CNS patients demonstrated an accumulation of CD68^+^CD163^+^ monocytes/macrophages and CD3^+^CD4^+^ and CD3^+^CD8^+^ T cells in the brain [8]. These recruited monocytes/macrophages expressed a M2 marker CD200R1, but not a M1 marker iNOS, suggesting a non-protective alternative activation of macrophages during PIIRS [8]. Accordingly, less than 1% of cryptococcal cells were located inside the monocytes/macrophages, which was indicative of a poor phagocytic function of monocytes/macrophages in the CNS [8].

Panackal et al. analyzed clinical samples of six previously healthy patients suffering from cryptococcal spinal arachnoiditis [33]. Although those patients had negative CSF fungal cultures after prior therapy with amphotericin B, they displayed significantly higher CSF levels of axonal damage biomarker NFL, soluble CD27 released by activated T cells, soluble CD21 released by B cells, associated with enhanced numbers of CSF cytotoxic NK cells, HLA-DR^+^ CD4^+^ T cells, HLA-DR^+^CD8^+^ T cells, B cells, monocytes, MyDCs, and plasmacytoid DCs, as compared to healthy donor controls [33]. These data suggest that PIIRS occurred among non-HIV-infected patients with cryptococcal spinal arachnoiditis.

## 5. Cryptococcal IRIS in Murine Models

Without immunosuppression, mice are naturally susceptible to *C. neoformans* infection and thus serve as model animals for HIV-associated cryptococcosis. The first animal model of Cryptococcal IRIS was developed by Eschke et al. through the reconstitution of CD4^+^ T cells in infected RAG-1^−/−^ mice [34]. The knockout mutation of RAG-1 gene results in lymphocyte-deficient mice, which simulates the deteriorating effect of HIV/AIDS on human immune competency. In this model, RAG-1^−/−^ mice were intranasally infected with 500 *C. neoformans* (serotype D strain 1841) and 32 days later injected intravenously with 2 × 10^6^ CD4^+^ T cells isolated from naïve mice [34] (Figure 1A). Following the adoptive transfer of CD4^+^ T cells, infected mice developed wasting and a systemic inflammatory disease including enhanced secretions of IFN-γ, IL-6, and TNF-α in the blood and granulomatous inflammation in the liver, compared to uninfected recipients and infected mice without CD4^+^ T cell adoptive transfer, demonstrating that reconstitution of CD4^+^ T cells induced IRIS-like symptoms in mice. Donor CD4^+^ T cells were more efficiently recruited to the brain of infected recipients and became more activated as assessed by their expression of CD69, compared with those of uninfected recipients. The adoptive transfer of CD4^+^ T cells led to the activation of microglial cells in the brain evidenced by increased size of microglial cell bodies and enhanced expression of MHC-II. Further experiments suggested that IFN-γ was produced by CD4^+^ T cells. IFN-γ produced by CD4^+^ T cells was required for the activation of microglial cells; however, it was not required for the activation of CD4^+^ T cells themselves and thus it is not required for the induction of *C. neoformans*-related IRIS. Interestingly, the systemic inflammation induced by reconstitution of CD4^+^ T cells did not affect fungal clearance in the brain as well as in the lung.

Khaw et al. established another mouse model of cryptococcal IRIS based on RAG-1^−/−^ mice [35]. In this model, RAG-1^−/−^ mice on C57BL/6 background were intranasally infected with 100 *C. neoformans* (serotype A strain H99). Purified CD4^+^ T cells (1 × 10^6^ cells per mouse) were adoptively transferred to the infected mice 3 weeks after infection through tail vein injection [35] (Figure 1B). Compared to infected mice without CD4^+^ T cell transfer, infected mice receiving CD4^+^ T cells exhibited significant weight loss and survived significantly shorter. In contrast, adoptive transfer of CD8^+^ T cells did not affect mouse survival. These observations demonstrated that reconstitution of CD4^+^ T cells but not CD8^+^ T cells induced cryptococcal IRIS in mice. IRIS mice displayed significantly higher serum levels of inflammatory cytokines including TNF-α and IL-6, enhanced numbers of myeloid cells in the lung, and accumulation of IFN-γ-secreting CD4^+^ T cells and higher numbers of myeloid cells including monocytes, neutrophils, and microglial cells in the brain. Interestingly, infected mice receiving wild-type CD4^+^ T cells survived significantly shorter compared to infected mice receiving IFN-γ^−/−^ CD4^+^ T cells, which was indicative of the pathological role of Th1 cells in IRIS. The reconstitution of CD4^+^ T cells did not affect fungal clearance but led to neurological damage with brain edema characterized by the enhanced expression of IFN-γ and aquaporin-4 (a critical protein regulating water flux in the brain [92,93]).

In contrast to intranasal infection, Neal et al. reported a murine model of cryptococcal IRIS and PIIRS based on intravenous infection [36]. Although cryptococcal infection starts in the lung, intravenous infections are the most relevant route of inoculation for brain infection experiments since *C. neoformans* escapes from the lung and gains access to the CNS from the blood. Intravenous inoculation eliminates the variable transmigration from the lung and mimics the process of fungal travel to the brain once the organism gains access to the blood. In this intravenous infectious model, C57BL/6 mice were inoculated with 1 × 10^6^ *C. neoformans* (serotype D strain 52D) and analyses were performed weekly after infection [36] (Figure 1C). Interestingly, the development of inflammatory responses in the brain occurred after the fungal growth plateau and was surprisingly correlated with neurological deterioration and mortality, demonstrating an exaggerated immune response in the CNS that was seen in cryptococcal IRIS and PIIRS patients. The inflammatory responses were characterized by a substantial recruitment of Ly6C^+^ myeloid cells and antigen experienced CD4^+^ and CD8^+^ T cells, with CD4^+^ T cells being predominant, high activation of microglial cells, and enhanced secretions of inflammatory chemokines and cytokines including CCL2, CCL3, TNF-α, and IFN-γ. Importantly, depletion of CD4^+^ T cells resulted in reduced IFN-γ production and lower numbers of CD8^+^ T cells and myeloid cells in the brain, and prevented the development of clinical symptoms and mortality despite a higher fungal burden in the brain, demonstrating that IFN-γ-producing CD4^+^ T cells promoted lethal immune responses in the brain despite fungal clearance [36]. It was further demonstrated that CXCL10, a ligand of CXCR3, was the most highly expressed chemokine detected in the brain of infected mice and that CXCR3^+^CD4^+^ T cells mediated neuronal damage, neurological deficits, and mortality of mice without a contribution to fungal clearance in this cryptococcal IRIS and PIIRS model [94]. Using this model, it was recently shown that CCR2 signaling promoted Ly6C^+^ inflammatory monocyte recruitment to the brain and indirectly modulated the accumulation of IFN-γ-secreting CD4^+^ T cells in the brain, leading to neuronal cell death and neuroinflammation, suggesting a therapeutic strategy aimed at inhibiting aberrant inflammation in IRIS and PIIRS patients by targeting the CCR2 pathway [95].

## 6. Concluding Remarks

*C. neoformans* escapes from the infected lung, enters the bloodstream, and then crosses the blood-brain barrier, colonizing the brain and causing brain infections [96,97,98]. Inflammatory responses, mainly a robust Th1 immune response, are required for cryptococcal clearance in the brain [10,11,12,99]. In HIV/AIDS patients, due to immune suppression, particularly quantitative and qualitative CD4^+^ T cell defects, limited inflammatory responses are induced, which facilitates fungal growth in the CNS. Uncontrolled growth of *C. neoformans* causes microbe-mediated brain damage [59]. ART efficiently restored cellular immune responses in HIV/AIDS patients co-infected with *C. neoformans*; however, it led to IRIS among one thirds of HIV/AIDS patients with cryptococcal meningoencephalitis. Likewise, PIIRS was also frequently seen in HIV-uninfected previously healthy individuals suffering from cryptococcal meningoencephalitis. Cryptococcal IRIS and PIIRS are aberrant inflammatory responses that lead to host-mediated brain damage [59].

Cryptococcal IRIS often occurred in patients with severe immunosuppression such as very low CD4^+^ T cell counts [15,24,26] and high fungal burdens [26,64] prior to ART. Due to depletion of CD4^+^ T cells in HIV/AIDS patients, fungicidal cells such as monocytes and macrophages have not received CD4^+^ T cell help via IFN-γ signaling to become fully activated [7]. This defect likely leads to higher fungal burdens in the brain due to inadequate killing and clearance of the fungal cells. Subsequent ART results in a rapid recovery of CD4^+^ T cells. Arguably, copious antigen presentation by antigen presenting cells to the recovered CD4^+^ T cells occurred, which induced excessive cellular immune responses, characterized by substantial recruitment of leukocytes to the CNS, and secretions of a large amounts of inflammatory cytokines including IFN-γ, TNF-α and IL-6 [30,32]. In line with clinical studies, animal experiments suggest that cryptococcal IRIS is induced by excessive activation of CD4^+^ T cells that secrete IFN-γ, with contribution of Ly6C^+^ inflammatory monocytes [34,35,36,94,95].

The mechanisms involved in cryptococcal IRIS still remain incompletely understood. To clear a cryptococcal infection in the CNS, inflammatory responses are definitively required; however, the immune responses restored by ART are obviously dysregulated. A recent study suggested that cryptococcal IRIS was associated with dysregulation of the IL-7/IL-7 receptor signaling pathway in T cell and monocyte activation [75]. However, many questions remain unanswered. For example, why is the recovery of immune responses dysregulated? Is there a mechanism balancing the protective immune responses and immunopathology? How are anti-inflammatory cytokines and regulatory T cells involved in the IRIS? Answering these questions will help our understanding of cryptococcal IRIS and PIIRS.

In summary, cryptococcal IRIS is a life-threatening complication. The biomarkers predictive of cryptococcal IRIS prior to ART and the biomarkers at the time of cryptococcal IRIS events have been well characterized in the past years. However, the mechanisms behand the dysregulation of the host immune responses are still largely unknown. Animal models of IRIS may provide a useful tool to address the mechanisms.

## Figures and Tables

**Figure 1 microorganisms-10-02419-f001:**
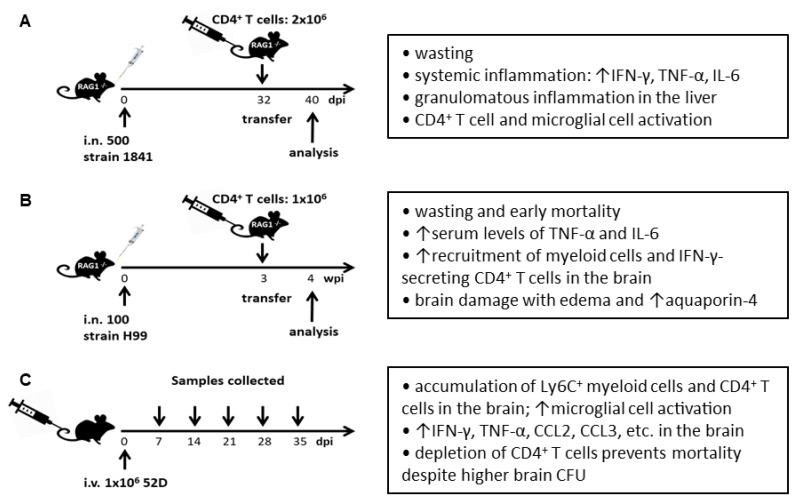
Graphic representation of murine models of cryptococcal IRIS. (**A**) RAG-1^−/−^ mice on C57BL/6 background were intranasally (i.n.) infected with 500 *C. neoformans* strain 1841 (Serotype D). 2 × 10^6^ CD4^+^ T cells isolated from naive mice were adoptively transferred to the infected RAG-1^−/−^ mice on day 32 post-infection (dpi) via tail vein injection. Analyses were performed on day 40 dpi [34]. (**B**) RAG-1^−/−^ mice on C57BL6 background were intranasally infected with 100 *C. neoformans* strain H99 (Serotype A). 1 × 10^6^ CD4^+^ T cells isolated from naïve mice were adoptively transferred to the infected mice on week 3 post-infection (wpi) via tail vein injection. Analyses were performed on 4 wpi [35]. (**C**) Wild-type C57BL/6 mice were intravenously (i.v.) infected with 1 × 10^6^ *C. neoformans* strain 52D (Serotype D). Samples were collected on day 7, 14, 21, 28 and 35 post-infection (dpi) [36].

**Table 1 microorganisms-10-02419-t001:** Biomarkers predictive of cryptococcal IRIS in HIV-infected patients prior to ART.

Sample Sources	Biological Activity	Biomarkers	References
**CSF**	Uncontrolled fungal growth	↑fungal burdens	[26,64]
	Poor inflammatory responses	↓protein levels	[63,64]
		↓leukocyte counts	[15,63]
		↓IFN-γ, TNF-α, IL-6, IL-8	[63]
	High CD8^+^ T cell response	↑CSF-blood ratio of CXCR3^+^CCR5^+^CD8^+^ T cells	[67]
	High monocyte/macrophage response	↑CCL2, CCL3, and GM-CSF	[15]
		↑CCL2/CXCL10 and CCL3/CXCL10 ratios	[67]
		↑CSF-blood ratio of CCL2, CCL3, and CXCL10	[67]
**Blood**	High fungemia	↑cryptococcal antigen titer	[24,69,70]
	Low Th1 responses	↓CD4^+^ T cell counts	[15,24,26]
		↓secretion of IFN-γ by peripheral CD4^+^ T cells; ↓activation of peripheral CD4^+^ T cells	[65,66]
		↓levels of TNF-α, G-CSF, GM-CSF, and VEGF; ↑CRP	[24]
	High Th2/Th17 responses	↑levels of IL-4 and IL-17	[24]
	High monocyte response	↑frequency of peripheral monocytes secreting TNF-α and IL-6	[68]
	Low anti-virus response	↓type I interferon gene expression	[71]
	Poor B cell function	↓IgM secretion in response to cryptococcal antigens	[72]

Abbreviations: ↑ increase; ↓ decrease; VEGF, vascular endothelial growth factor; CRP, C-reactive protein.

**Table 2 microorganisms-10-02419-t002:** Biomarkers at the time of cryptococcal IRIS events.

Sample Sources	Biological Activity	Biomarkers	References
**CSF**	High T cell responses	↑CD4^+^ T cells	[23,73]
	High inflammatory cytokines	↑IFN-γ, TNF-α, G-CSF, VEGF, eotaxin	[63]
	High monocyte/macrophage activation	↑monocytes	[73]
		↑sCD14 and sCD163	[74]
	High NK cell activation	↑PD-L1 expression on NK cells	[73]
**Blood**	High monocyte activation	↑CD14^+^CD86^+^ or CD14^+^HLA-DR^+^ cells	[75]
	High CD4^+^ T cell activation	↑IL-2^+^/IL-17^+^ CD4^+^ T cells	[66]
	High inflammatory cytokines	↑IL-6 and CRP	[24]
		↑IFN-γ and IL-6	[79]
	Compensatory anti-inflammation	↑IL-10	[79]
		↑Foxp3^+^CD4^+^ T cells	[81,82]

Abbreviations: ↑, increase; VEGF, vascular endothelial growth factor; sCD14, soluble CD14; sCD163, soluble CD163; PD-L1, programed death ligand 1; CRP, C-reactive protein.

**Table 3 microorganisms-10-02419-t003:** Biomarkers at the time of cryptococcal PIIRS events.

Sample Sources	Biological Activity	Biomarkers	References
**CSF**	Axonal damage	↑NFL	[8,33]
	High T cell and B responses	↑CD4^+^ and CD8^+^ T cells, B cells	[8,33]
	High inflammatory cytokines and chmokines	↑IFN-γ, IL-6, IL-18, CXCL10, MCP-1, MIP-1α, MCP-3, and MCP-3β	[8]
	Low Th2 cytokines	↓IL-4	[8]
	High innate immune cell responses	↑NK cells, monocytes, and MyDCs	[8,33]
	Compensatory anti-inflammation	↑IL-10 and IL-13	[8]
**Blood**	High T cell activation	↑CD4^+^ and CD8^+^ T cells secreting IFN-γ	[8]

Abbreviations: ↑, increase; ↓, decrease; NFL, neurofilament light chains.

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
