# Peer review of "Cryptococcal Immune Reconstitution Inflammatory Syndrome: From Clinical Studies to Animal Experiments"

_microorganisms, 2022, doi:10.3390/microorganisms10122419_

Round 1

Reviewer 1 Report

In general, this is a very well written and comprehensive review in the area of cryptococcal IRIS, providing thorough review of the topic and fairly presenting literature and our state of knowledge in this field. The authors should be congratulated for their hard work.

I have only a few minor suggestions for improvement.

1) The review in essence describes 2 syndromes IRIS and PIIRS, but the title mentions only IRIS. Further, it is not easy to grasp reading this review how PIIRS is different from IRIS. Is this the same outcome in 2 subsets of patients or two different syndromes that have an excessive inflammation as a common denominator? It would definitely help to provide an additional table comparing and contrasting IRIS and PIIRS, at least in areas when readouts have been studied for both. This would help readers to reach conclusion if PIIRS is most likely the same with IRIS with the exception of the "reconstitution part", or they are related but still different enough to be consider as separate disorders. If the latter, perhaps PIIRS should be also included in the title of this review.

2. In several areas review articles are cited and not the primary sources. If the authors wish to cite a review, they should add "reviewed in..." when citing, to indicate that they agree with the previously formulated opinion based on other reviews. However, for some of the statements it would be better to obtain the support from original primary sources.

3. The sentence ending pg 1- starting pg 2 is a bit vague, saying "it is believed that". The role of IFN-g in clearance has been firmly established across dozens of studies, both models and the clinical data, so it is known. The role of the Th2 is less affirmed, since mouse models using naive SPF animals are capable to demonstrate extremes of immune polarization that are very rarely seen in human patients and the data are not always the same. It would be good to be more specific about it right up front.

4. Pg 2. Ln 10. in Part 2. I would suggest changing word "demonstrating" to "supporting." These cited association studies "support" but not demonstrate that certain cytokines promote clearance. 

5. Part 6. Concluding remarks. This part mostly summarizes beginning of the article, but it could contain a better synthesis of the whole review, bulleting the most important "take home messages".  What parts of our understanding is firm and what parts are still controversial? Are the existing studies and models sufficient to dissect the existing questions, what are the other data and models needed to advance the therapy? These ideas start to be formulated but could be stated with more confidence and clarity.

Author Response

In general, this is a very well written and comprehensive review in the area of cryptococcal IRIS, providing thorough review of the topic and fairly presenting literature and our state of knowledge in this field. The authors should be congratulated for their hard work.

I have only a few minor suggestions for improvement.

  • The review in essence describes 2 syndromes IRIS and PIIRS, but the title mentions only IRIS. Further, it is not easy to grasp reading this review how PIIRS is different from IRIS. Is this the same outcome in 2 subsets of patients or two different syndromes that have an excessive inflammation as a common denominator? It would definitely help to provide an additional table comparing and contrasting IRIS and PIIRS, at least in areas when readouts have been studied for both. This would help readers to reach conclusion if PIIRS is most likely the same with IRIS with the exception of the "reconstitution part", or they are related but still different enough to be consider as separate disorders. If the latter, perhaps PIIRS should be also included in the title of this review.

Response: IRIS is common in HIV-patients co-infected with C. neoformans following ART. PIIRS occurs in previously healthy individuals with cryptococcal meningitis and is rare. Yes, PIIRS is very similar to IRIS except the “reconstitution part”. We added a table (Table 3) to summarize the biomarkers at the time of PIIRS event.

  1. In several areas review articles are cited and not the primary sources. If the authors wish to cite a review, they should add "reviewed in..." when citing, to indicate that they agree with the previously formulated opinion based on other reviews. However, for some of the statements it would be better to obtain the support from original primary sources.

Response: thanks, we added “reviewed in…” accordingly.

  1. The sentence ending pg 1- starting pg 2 is a bit vague, saying "it is believed that". The role of IFN-g in clearance has been firmly established across dozens of studies, both models and the clinical data, so it is known. The role of the Th2 is less affirmed, since mouse models using naive SPF animals are capable to demonstrate extremes of immune polarization that are very rarely seen in human patients and the data are not always the same. It would be good to be more specific about it right up front.

Response: we removed “it is believed”.

  1. Pg 2. Ln 10. in Part 2. I would suggest changing word "demonstrating" to "supporting." These cited association studies "support" but not demonstrate that certain cytokines promote clearance. 

Response: we changed “demonstrating” to “supporting”.

  1. Part 6. Concluding remarks. This part mostly summarizes beginning of the article, but it could contain a better synthesis of the whole review, bulleting the most important "take home messages".  What parts of our understanding is firm and what parts are still controversial? Are the existing studies and models sufficient to dissect the existing questions, what are the other data and models needed to advance the therapy? These ideas start to be formulated but could be stated with more confidence and clarity.

Response: thanks for the valuable comments. We were trying to make a “conclusion” paragraph to highlight the major points we described in the whole manuscript. We do agree that the manuscript will benefit from “take home message”. Thus, in the revised version we added a few sentences to state “what have known and what we do not know” at the end of the manuscript.

Reviewer 2 Report

The manuscript is very well written and extremely pleasant to read.

I missed an introduction about the microorganism. Authors should comment on Cryptococcus and the polysaccharide capsule. Mainly because of the immunomodulatory capacity of the polysaccharide components that are well described in the literature (glucuronoxylomannan and galactoxylomannan

A review on cryptococcosis is unacceptable without discussing the fungus and especially the capsule

Below are some references that the authors could use in the manuscript:

a) Immunomodulatory Role of Capsular Polysaccharides Constituents of Cryptococcus neoformans. Decote-Ricardo D, LaRocque-de-Freitas IF, Rocha JDB, Nascimento DO, Nunes MP, Morrot A, Freire-de-Lima L, Previato JO, Mendonça-Previato L, Freire-de-Lima CG. Front Med (Lausanne). 2019 Jun 19;6:129. doi: 10.3389/fmed.2019.00129.

b) The capsule of Cryptococcus neoformans.

Casadevall A, Coelho C, Cordero RJB, Dragotakes Q, Jung E, Vij R, Wear MP. Virulence. 2019 Dec;10(1):822-831. doi: 10.1080/21505594.2018.1431087.

c) Unraveling Capsule Biosynthesis and Signaling Networks in Cryptococcus neoformans. Jang EH, Kim JS, Yu SR, Bahn YS.

Microbiol Spectr. 2022 Oct 26:e0286622. doi: 10.1128/spe

d) Extracellular vesicles from Cryptococcus neoformans modulate macrophage functions.Oliveira DL, Freire-de-Lima CG, Nosanchuk JD, Casadevall A, Rodrigues ML, Nimrichter L.

Infect Immun. 2010 Apr;78(4):1601-9. doi: 10.1128/IAI.01171-09.

e) Capsular polysaccharides from Cryptococcus neoformans modulate production of neutrophil extracellular traps (NETs) by human neutrophils. Rocha JD, Nascimento MT, Decote-Ricardo D, Côrte-Real S, Morrot A, Heise N, Nunes MP, Previato JO, Mendonça-Previato L, DosReis GA, Saraiva EM, Freire-de-Lima CG. Sci Rep. 2015 Jan 26;5:8008. doi: 10.1038/srep08008.

Author Response

The manuscript is very well written and extremely pleasant to read.

I missed an introduction about the microorganism. Authors should comment on Cryptococcus and the polysaccharide capsule. Mainly because of the immunomodulatory capacity of the polysaccharide components that are well described in the literature (glucuronoxylomannan and galactoxylomannan

A review on cryptococcosis is unacceptable without discussing the fungus and especially the capsule

Below are some references that the authors could use in the manuscript:

  1. a) Immunomodulatory Role of Capsular Polysaccharides Constituents of Cryptococcus neoformans. Decote-Ricardo D, LaRocque-de-Freitas IF, Rocha JDB, Nascimento DO, Nunes MP, Morrot A, Freire-de-Lima L, Previato JO, Mendonça-Previato L, Freire-de-Lima CG. Front Med (Lausanne). 2019 Jun 19;6:129. doi: 10.3389/fmed.2019.00129. 
  2. b) The capsule of Cryptococcus neoformans.

Casadevall A, Coelho C, Cordero RJB, Dragotakes Q, Jung E, Vij R, Wear MP. Virulence. 2019 Dec;10(1):822-831. doi: 10.1080/21505594.2018.1431087. 

  1. c) Unraveling Capsule Biosynthesis and Signaling Networks in Cryptococcus neoformans. Jang EH, Kim JS, Yu SR, Bahn YS.

Microbiol Spectr. 2022 Oct 26:e0286622. doi: 10.1128/spe 

  1. d) Extracellular vesicles from Cryptococcus neoformans modulate macrophage functions.Oliveira DL, Freire-de-Lima CG, Nosanchuk JD, Casadevall A, Rodrigues ML, Nimrichter L.

Infect Immun. 2010 Apr;78(4):1601-9. doi: 10.1128/IAI.01171-09. 

  1. e) Capsular polysaccharides from Cryptococcus neoformans modulate production of neutrophil extracellular traps (NETs) by human neutrophils. Rocha JD, Nascimento MT, Decote-Ricardo D, Côrte-Real S, Morrot A, Heise N, Nunes MP, Previato JO, Mendonça-Previato L, DosReis GA, Saraiva EM, Freire-de-Lima CG. Sci Rep. 2015 Jan 26;5:8008. doi: 10.1038/srep08008.

Response: thanks for the comment and we agree with the reviewer. We have added a short statement (“However, C. neoformans possesses multiple virulence factors, mainly the polysaccharide capsule [reviewed in 19,20], that is capable of modulating host immune responses [21,22].”) at the end of the first paragraph in page 2.